# Sulfide Globule and a Localized Domain Ultra-Enriched in PGMs in the Main Reef Anorthosite from the Yoko-Dovyren Massif

**Ivan V. Pshenitsyn** [1,*], **Alexey A. Ariskin** [1,2], **Dmitry V. Korost** [2], **Sergei N. Sobolev** [1], **Vasily O. Yapaskurt** [2] **and Georgy S. Nikolaev** [1]

1   Vernadsky Institute of Geochemistry and Analytical Chemistry, Russian Academy of Sciences, Kosygin Str. 19, 119991 Moscow, Russia; ariskin@rambler.ru (A.A.A.); ssn_collection@bk.ru (S.N.S.); gsnikolaev@rambler.ru (G.S.N.)
2   Faculty of Geology, Lomonosov Moscow State University, Leninskie Gory 1, 119234 Moscow, Russia; dkorost@mail.ru (D.V.K.); yvo72@geol.msu.ru (V.O.Y.)
*   Correspondence: lotecsi@gmail.com; Tel.: +7-9169765141

**Abstract:** The results of a detailed examination of an anomalously PGM-rich anorthositic fragment from the Main Reef of the Yoko-Dovyren massif (Northern Transbaikalia, Russia) are presented. This fragment is to represent a 15 mm core drilled out from a typical low-sulfide PGE-rich anorthosite, occurring within the transition zone between troctolite and a rhythmically stratified sequence of olivine gabbro. Coupling multistage X-ray computed tomography (CT) with SEM studies allowed for revealing a heterogeneous distribution of PGMs and sulfides observable as (i) the main 4 mm sulfide globule containing some small PGMs around its periphery, with (ii) the bulk of the PGMs concentrated within a 3 mm sized scattered sulfide nest, comprising about 6 vol.% of the globule and located at a distance of 2–3 mm from it. Mass-balance calculations showed that the average sulfide composing this nest is 120fold richer in PGE than the sulfide globule. Calculations of sulfide minerals proportions showed that the globule consists of 39 vol.% Po, 21% Pn, 34% Cub, and 6% Ccp (consistent with 35.2 wt.% S, 48.2% Fe, 6.4% Ni, 9.9% Cu, and 0.4% Co), whereas the PGM-enriched sulfide domain includes (vol.%): Po—34, Pn—15, Ccp—23, and Cub—28 (respectively, S—35.2 wt.%, Fe—45.8%, Ni—4.6%, Cu—14.2%, and Co—0.3%). Thus, the PGM-enriched nest demonstrates an obvious increase in Cu relative to the sulfide globule. Further SEM studies of four thin sections of the globule and associated nest showed that they differ not only in the ratios of base metal sulfides, but also in the PGE mineralogy. The globule contains more high-temperature PGMs, such as moncheite, while the nest is enriched in "low-temperature" PGMs, including notable amounts of lead and mercury. The overwhelming majority of the numerous PGMs in the unusual domain were detected as tetraferroplatinum, with subordinate potarite and zvyagintsevite, associated with chlorite and apatite. Such a subdivision of anorthositic sulfides into two types demonstrating different composition and mineralogy, as well as contrasting distributions of PGE in the sulfide segregations, was established for the first time! The origin of the contrast PGM-sulfide assemblages is discussed.

**Keywords:** Dovyren; PGE-rich anorthosite; computed tomography; sulfides; platinum group elements

## 1. Introduction

This research is a continuation of petrological studies targeted at the search for PGMs by X-ray computed tomography (CT) and their detailed characterization by SEM in low-sulfide PGE-rich anorthosite from the Yoko-Dovyren massif (Northern Transbaikalia, Russia) [1]. The previous publication was focused on methodological aspects of the CT-examinations, as well as general features of the distribution of PGMs in most sulfide mineralized areas of the PGE-anorthosite. Herein, we present more localized and somewhat unusual observations, which, probably, make further insight into the processes responsible for the formation of low-sulfide reef-style mineralization.

Increasingly, CT methods are being used to visually and numerically characterize silicate–sulfide relationships in mineralized rocks of layered intrusions [2–7]. An obvious disadvantage of classical petrographic methods is the extrapolation of 2D information to 3D structures, which can lead to various distortions. Thus, slides made from a random rock slice can result in a false idea of important parameters that are necessary for understanding the formation processes for magmatic sulfides and related PGMs. The major aim of CT studies includes obtaining statistically based quantitative information on the distribution of the mineral phases with different X-ray absorptions, which seems to generate insight into the mechanisms of their genesis. Thus, on the basis of the morphological analysis of sulfides and their connectivity, the authors of [4] made a conclusion about the probable percolation of immiscible sulfide liquid in olivine cumulates from komatiite flows of the Norsman-Viluna greenstone belt in Western Australia. This is consistent with an experimental study on the wetting behavior of minerals in sulfide–silicate systems [8]. In [3], using the CT-approach, authors concluded that there were several generations of sulfides in olivine cumulates: some of them originated in situ, while others were the result of percolation and accumulation. A significant part of the cited study was focused on statistical data on the size distribution of sulfide globules, as well as on the degree of their "sphericity". In a fundamental review [9], the authors discussed the results of CT studies of disseminated sulfide mineralization from the Sunrise Dam gold mine in Eastern Australia and from the Mount Kit Cu-Ni deposit, focusing on the "CSD" of sulfides in terms of their equivalent sphere diameter. This parameter is useful for the statistical comparison of ensembles of individual grains that have slightly different shapes.

Based on a petrographic and CT examination of samples from the Merensky Reef (Bushveld Complex, South Africa), it was concluded that the lower chromitite layer of the MR may play a crucial role as a physical barrier that prevents the sulfide liquid percolation from above, thus leading to its accumulation [5]. The low degree of sulfide connectivity in chromitite interlayers (in contrast to MR melanorite), and the overall distribution of sulfide segregations within the Merensky Reef is consistent with the experimental studies on the wetting of minerals in sulfide–chromite–silicate systems [10]. The other studies were focused on the distribution of platinum group minerals in the Merensky Reef [9]. A number of important observations have been made combining data on the size, shape, quantities, and spatial association of PGMs with chromite, sulfides, and silicates. In particular, based on the similarity of these characteristics for PGMs from the upper and lower chromitite layers in the MR, it was concluded that the formation of two border chromitite layers did not affect PGMs, which originated as a result of a long-standing evolution of the original sulfide liquids, including both their crystallization history and, probably, post-cumulus late-stage processes [6]. Other groups of researchers came to similar conclusions after CT investigations of the low-sulfide rocks from the Platinova reef in the Skaergaard intrusion [7,11]. In addition, the quantitative parameters which resulted from the CT studies (see above) allow one to better understand the possible fluid dynamics of sulfide-saturated magmas and the dynamics of the transfer of sulfide blebs through the cumulate piles.

Thus, different shapes of the reconstructed "CSD" for sulfide globules, in combination with computer simulations, make it possible to propose a generalized genetic scheme of the formation and disintegration of sulfides depending on the style of magma dynamics (laminar or turbulent) [12].

Pshenitsyn et al. [1] presented the first multi-scale results of CT examinations of a PGE-anorthosite from the Dovyren intrusion (sample 13DV539-9). These results provided valuable information on the contents and connectivity of the anorthositic sulfides, as well as the distribution of high X-ray absorption phases as probable noble metals minerals. These findings were supplemented by mineralogical studies of the two largest grains, including a 30 μm size moncheite and 120 μm size electrum. Based on our experience, CT imaging should precede every study of PGE mineralization associated with magmatic sulfides, if one plans to conduct detailed petrological studies to provide conclusions regarding the origin of the PGMs occurrences. Of particular interest is that such an approach combining the use of scanning electron microscopy, classical optical petrography, and CT imaging allows for the first-order estimation of the total concentration of PGE in the bulk sulfide, which can be compared with calculations of 100% sulfide composition from the whole-rock PGE analyses. A problem is that despite the demonstrated efficiency of the combined CT-approach, the lack of sufficient data that would statistically link findings of PGMs with their compositions in the paper [1] did not allow for well-argued conclusions on probable mechanisms of the formation of the noble-metal mineralization in the low-sulfide anorthosites.

## 2. Geological Background

The layered Yoko-Dovyren mafic-ultramafic intrusion (hereinafter Dovyren) is hosted in the Baikalides of Northern Transbaikalia [13–18] and has size of 26 × 3.5 × ~5 km. It belongs to the Synnyr-Dovyren volcano–plutonic complex, with—728.4 ± 3.4 Ma for the Dovyren intrusion and 722 ± 7 Ma for the associated rhyolites [18]. Its cumulate succession in the thickest central and most differentiated part is composed of the near-contact zone olivine picrodolerite, followed with plagiolherzolites and the dunite zone (about one third of the cross section) transitioning to troctolites (with anorthositic veins and schlieren), and the upper strata of olivine to olivine-free gabbronorites and pigeonite gabbro [16–18]. Various types of sulfide mineralization occur in the Yoko-Dovyren massif, including (i) massive Cu-Ni ores of the Baikal deposit, which are hosted by melanocratic olivine gabbronorite composing underlying sills and apophyses (subparallel to the main body), (ii) disseminated mineralization in dunites, (iii) low-sulfide and PGE mineralization in the so-called Konnikov zone [19], as well as (iv) PGE-poor, low-copper mineralization in gabbroids near the roof of the Dovyren intrusion.

However, many Russian specialists focused their attention on the so-called Main Reef [18,19], or Reef I [20–22], which has been mapped as a texturally taxitic feldspar-rich horizon within the variable thick transition zone between typical troctolites and olivine gabbro (see Figure 1a). Following from some indirect analogies with PGE-rich horizons of the Bushveld Complex, the whole taxitic zone is considered by some authors as a "Critical Zone" [21], which in its upper part includes a number of randomly distributed schlierens and lenses of PGE-rich anorthosite, up to several meters thick (Figure 1b). These anorthositic bodies associate with gabbro-pegmatites and are hosted by mesocratic to leucocratic troctolites (Figure 1c). The PGE mineralization detected here was referred to the so-called "Stillwater type" [20,21], as up to 35 PGE minerals were found, with moncheite, kotulskite, and other bismuthtellurides being predominant, and other PGMs such as tetraferroplatinum, potarite, zvyagintsevite, and atokite occurred. Due to the rather uneven distribution of sulfides in the anorthosite, the total concentration of noble metals

and PGE in the rocks varies from 0.3 to ~6 g/t, but the bulk contents of about 4–5 g/t prevail [20–22].

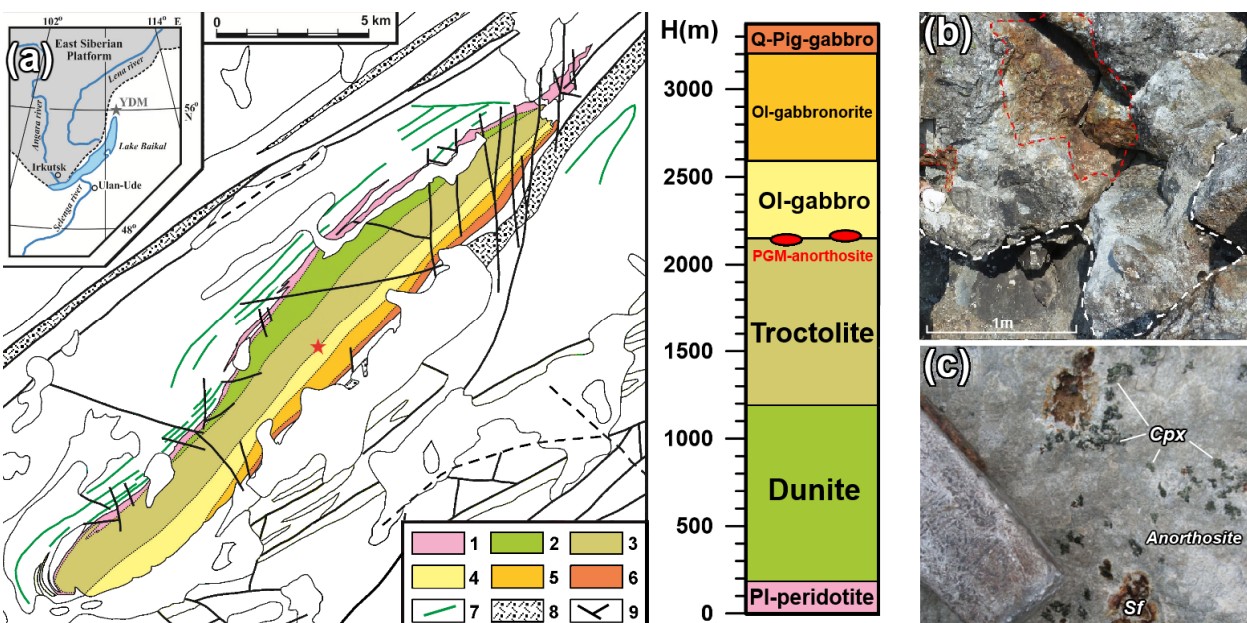

**Figure 1.** Schematic map of the Yoko-Dovyren massif, general stratigraphic succession, and examples of low-sulfide PGE-rich anorthosites: (**a**) modified after [1], where (1) plagioperidotite, (2) plagiodunite and dunite, (3) troctolite, (4) olivine gabbro, (5) olivine gabbronorite, (6) quartz- and pigeonite-bearing gabbronorite, (7) mafic to ultramafic sills and dykes, (8) high-Ti basaltic flows, and (9) faults. The red star marks the location of the outcrop of the studied anorthosite; (**b**) exposure of the anorthositic schlieren (white dotted line) with isolations of low-sulfide mineralization rich in PGE minerals (red dotted line); and (**c**) poikilitic grains of clinopyroxene (*Cpx*) in anorthosite hosting nests of sulfides (*Sf*) substituted by iron hydroxides.

The geological forms of ore-bearing anorthosites are different: first of all, they occur as schlieren-like and lens-shaped isolations with a thickness from the first cm to a meter and more, varying in composition from almost pure feldspar rocks to gabbroanorthosites, less often with the fringing of gabbro-pegmatite, taxitic troctolite, and olivine leucogabbronorite (Figure 1b). Along the strike, they may extend for 2–5 m, rarely 40 or more meters, forming a series of discontinuous, clumped ore occurrences that can be traced for more than 20 km [22,23]. Often, they are oriented crosswise to the strike of the Dovyren massif and cut the stratigraphic rock sequence.

The composition of plagioclase in anorthosites corresponds to bitownite and varies in the range of 82.3%–87.5% An [18,21], sometimes with rims of more calcium plagioclase containing 88%–90% An [1]. These rocks typically contain 1 to 5% ortho- (En 73.5–78.2) and clinopyroxene (En 39.6–44.3) [21] (Figure 2d–f). Both Opx and Cpx may occur as large poikilitic crystals observed in thin sections as pyroxenes occupying the pore (interstitial) space (Figure 2d–f); however, only orthopyroxene composes the reaction rims (Figure 2f) around separate olivine grains. Olivine is the third by prevalence but genetically important mineral, occurring as individual grains up to 0.5 mm in size or clusters of the olivine crystals, mostly containing Fo 83%–85% Fo (Figure 2c,f). Pyrrhotite, pentlandite, cubanite, and chalcopyrite are the main base metal sulfides, which form mostly separate domains of a net texture hosted by plagioclase crystals (Figure 2d,e). In many places, those domains look similar to the intercumulus pyroxenes (Figure 2a,b). The average size of the disseminated sulfide grains is about 0.5 mm, but segregations up to first millimeters may also occur (Figure 2a). Bulk sulfide contents in the mineralized anorthosites are variable, but, as a

rule, do not exceed 4–5 wt.%. In places of the increased amount of sulfide, plagioclase is subjected to more extensive secondary alteration, with the formation of clinozoisite and prehnite. Correspondingly, a low temperature assemblage of clinozoisite, chlorite, albite, prehnite, and phlogopite may develop [1,22].

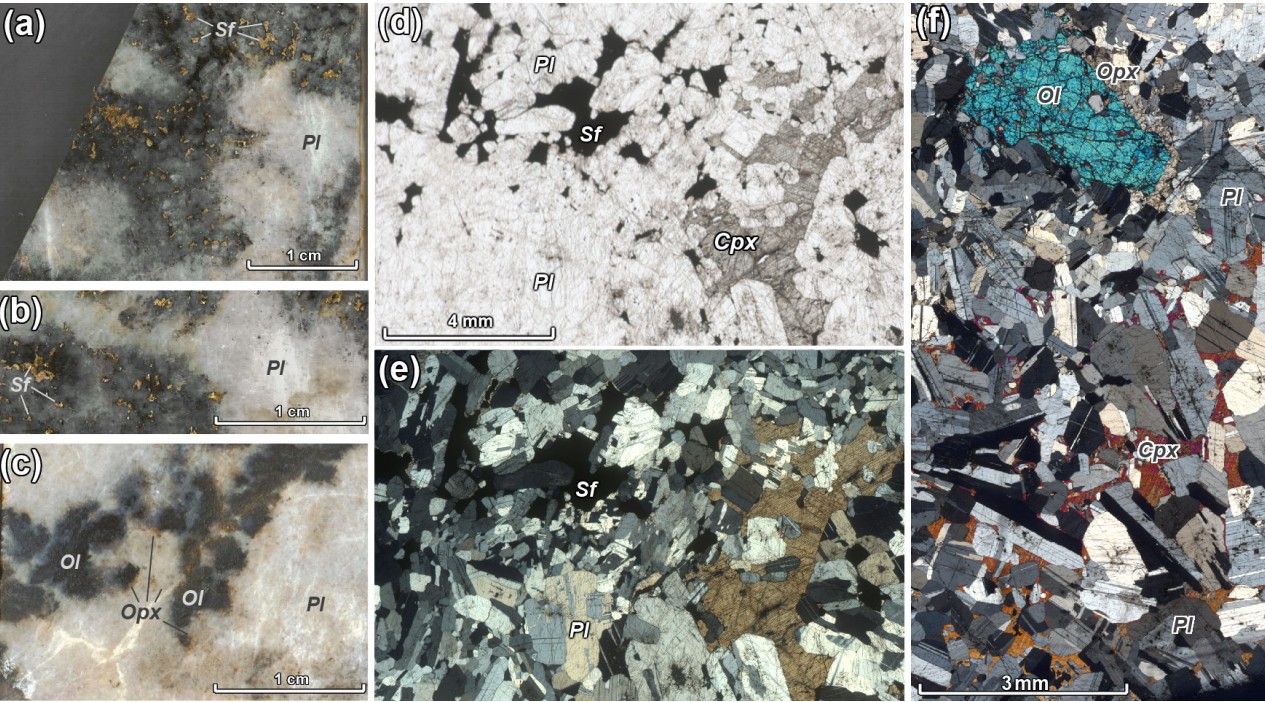

**Figure 2.** Images of fragments of the examined anorthositic block referred as DV653: (**a**–**c**) polished sections of the mineralized sample DV653-5, (**d**,**e**) panoramas of thin sections of DV653-7 hosting disseminated sulfides (crossed and parallel analyzers), and (**f**) section of the DV653-5 anorthosite with poikilitic clinopyroxene and olivine phenocryst with the reaction Opx rim. *Sf*—sulfide, *Pl*—plagioclase, *Ol*—olivine, *Opx*—orthopyroxene, and *Cpx*—clinopyroxene.

## 3. Materials and Methods

We focused on investigations of the anorthositic sample DV653-5 (Figure 3) as a piece from a large block DV653 sampled from the same outcrop as the sample 07DV146-1 and its fragments (AA06a-1 and AA06a-2), which have been presented in [24]. In fact, DV653-5 is about 5 × 7 × 15 cm in size, being a typical low-sulfide PGE-rich anorthosite with irregularly shaped elongated areas of sulfides, zones of poikilitic pyroxene (Figure 2d,e), and inclusions of olivine aggregates (Figures 2a–c and 3a). Unlike the sample 13DV539-9 from another anorthositic schlieren, which has a banded distribution of disseminated sulfides [1], DV653-5 has several irregularly shaped areas of similar mineralization (see Figures 2a–c and 3a). X-ray computed tomography was performed in two stages using two instrumental systems, which provided data of different informativeness and resolution. At the first stage, the whole sample was scanned using an RKT-180 tomograph (produced by Geologika, Novosibirsk, Russia) with a resolution of 100 μm. Then, two 10 and 15 mm diameter cylinders were drilled out to sample areas enriched in sulfide domains (Figure 3a), with the 15 mm core capturing the 4 mm sulfide globule detected during the previous CT imaging. The 10 and 15 mm cores were imaged using a SkyScan-1172 scanner (Bruker, Germany) to obtain data stacks with a resolution of about 10 μm, and, in the case of an additional smallest core of 3 mm in diameter, with a resolution of 3 μm.

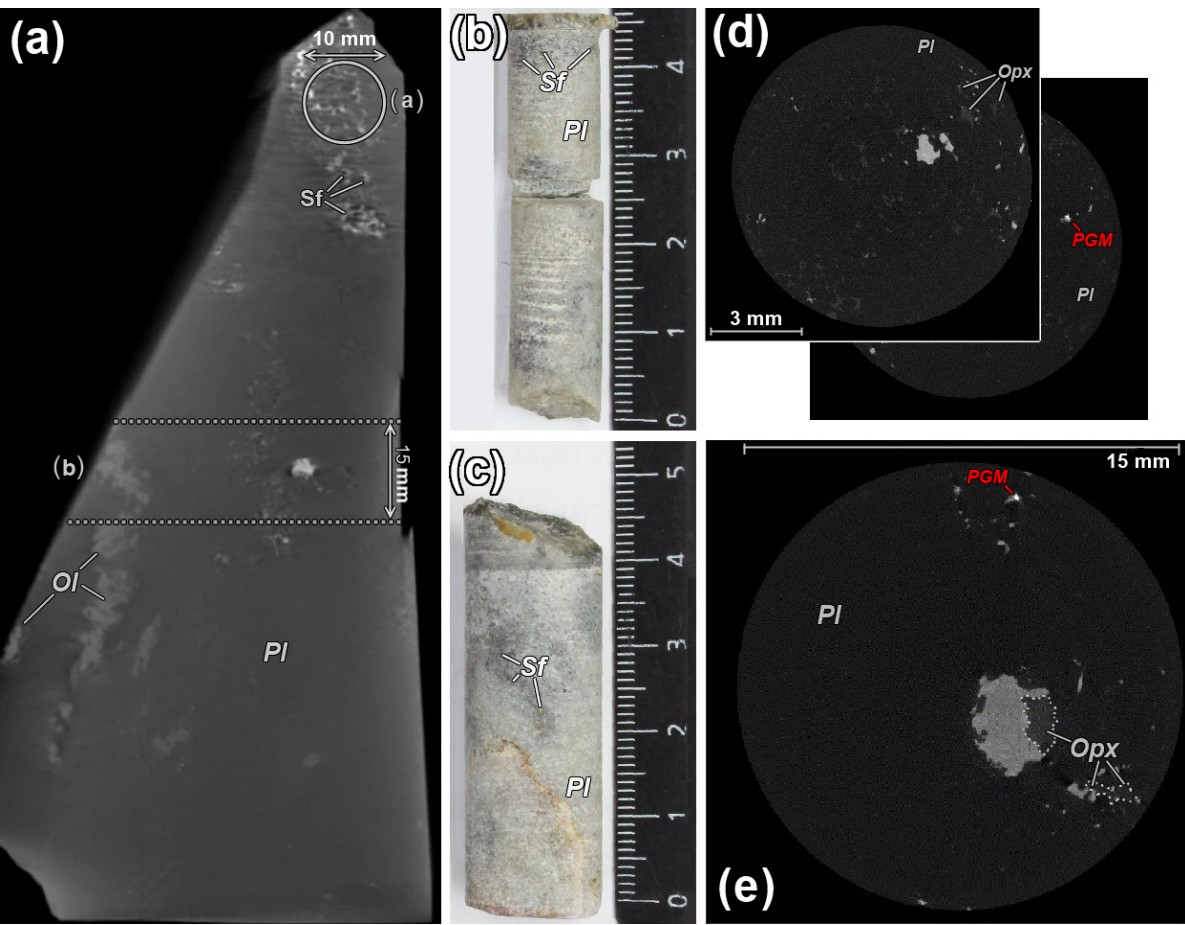

**Figure 3.** Studied areas in the DV653-5 anorthosite: (**a**) results from the first stage of the low-resolution scanning, where lighter clusters of olivine and disseminated sulfides are clearly distinguished in the gray plagioclase matrix (the circle marks the location of the 10 mm core drilled across the sample), (**b**,**c**) are 10 and 15 mm cores, respectively, the dark areas correspond to sulfide clusters in the rock, (**d**) examples of several STvox "frames" from the 10 mm core data stack, and (**e**) an example of an X-ray dense section of the 15 mm core with a 4 mm sulfide globule at the center, in association with putative orthopyroxene and associated interstitial sulfides.

Microprobe studies of mineral compositions were carried out in the Laboratory of Local Methods of Studying Matter (the Faculty of Geology, Lomonosov MSU, Moscow, Russia) using a JSM-6480LV electron microscope with a tungsten thermionic cathode equipped with an X-Max-N50 energy dispersive spectrometer (Oxford Instruments, Abingdon, UK). The standards and samples were measured in the focused probe mode at an accelerating voltage of 20 kV and a probe current of 10 nA. In this case, the standards for metals, stoichiometric oxides, and sulfides were used. The INCA shell (Oxford Instruments, version 21b) was used to process the measurement results by the XPP-correction algorithm, which ensured the accuracy of the content estimation for the main elements in the range of 0.5–2 oct.%.

The petrographic description and microphotographs were taken using the Altami MET 1C (Altami, Saint Petersburg, Russia) and CARL ZEISS AXIOLAB.A1 (ZEISS Russia & CIS, Moscow, Russia) optical microscope at the Faculty of Geology (Lomonosov MSU, Moscow, Russia).

## 4. Results

### 4.1. X-Ray Computed Tomography

The specification of X-ray contrast phases in the 15 mm core demonstrated that silicate minerals (plagioclase and pyroxenes) make up 98.86% of the rock volume. Sulfides occupy practically the whole rest of the volume, and the volume concentration of potential PGMs (hereinafter referred to as "p.PGM") is not higher than 0.001% (Table 1). At the same time, due to the presence of one relatively large sulfide globule, in which most of the sulfides are concentrated (Figure 4), their bulk connectivity in the sample reaches as much as 70%. Note that the term "connectivity" in CT studies is attributed to a fraction of the largest coherent object in the overall volume of a given phase. The CT data for the 10 mm core display a more uniform distribution of sulfides over the plagioclase–pyroxene matrix. In addition, in Table 1, data for two parts of the 15 mm core are given, which represent two virtual sections, including (i) the globule separately (Figure 4b, 15 mm (A)) and (ii) the adjacent associated domain of sulfides anomalously enriched in p.PGMs (Figure 4c,d, (15 mm (B)).

**Table 1.** Content of X-ray contrast phases in the volume of 15 and 10 mm cylinders, as well as individual sections of the 15 mm cylinder (15 mm (A)—globule; 15 mm (B)—p.PGM enriched nest).

| Sample | Silicates Content, vol.% | Sulfides Content, vol.% | Sulfides Connectivity, % | p.PGMs Numbers (vol, %) | p.PGM Maximum Linear Diameter (μm) |
|---|---|---|---|---|---|
| DV653-5 | | | | | |
| 653-5—10 mm | 97.56 | 2.44 | 25 | 65 (0.001) | 60 |
| 653-5—15 mm | 98.86 | 1.14 | 70 | 71 (0.001) | 450 |
| 15 mm (A) | 95.79 | 4.21 | 100 | 18 (0.0001) | 35 |
| 15 mm (B) | 99.32 | 0.68 | 19 | 45 (0.01) | 450 |

A detailed examination of the three-dimensional reconstructions of the sample (Figure 4a) shows that a sphere-like globule with a diameter of about 4 mm (Figure 4b) is clearly distinguished among a cloud of finely disseminated sulfides, which contain only a few grains of p.PGMs. At the same time, it is associated with a domain of sulfides anomalously enriched in p.PGMs (Figure 4c,d), at a distance of few mm from the globule. Note that a much larger volume of disseminated sulfides (5.7-fold with respect to the enriched nest) is observed in the vicinity and above the globule (see Figure 4a), with only a few very small grains of p.PGMs (Figure 4a).

The strategy of the following studies included two main stages: (1) the discovered globule was cut on several microslides considered as reflecting its different "hypsometric levels" to be used for further petrological and mineralogical investigations, including the assessment of its "average sulfide composition", and (2) the second stage included the stepwise sawing off the area of the p.PGM-rich sulfide nest, with the purpose to uncover and to study as many p.PGM grains as possible. One can visualize this as an anti-3D printer.

Such an invasive method was chosen as a large number of objects of interest were found in the very small volume, so that the preparation of checkers or slides would allow us to obtain only a couple of thin sections to study. This approach allowed for evaluating a representative number of PGM compositions, favoring calculations of the "average sulfide composition" in the unusual sulfide domain. In addition, direct comparisons and correlations of the morphological features of PGMs obtained by the CT-approach with the results of SEM microscopy became possible.

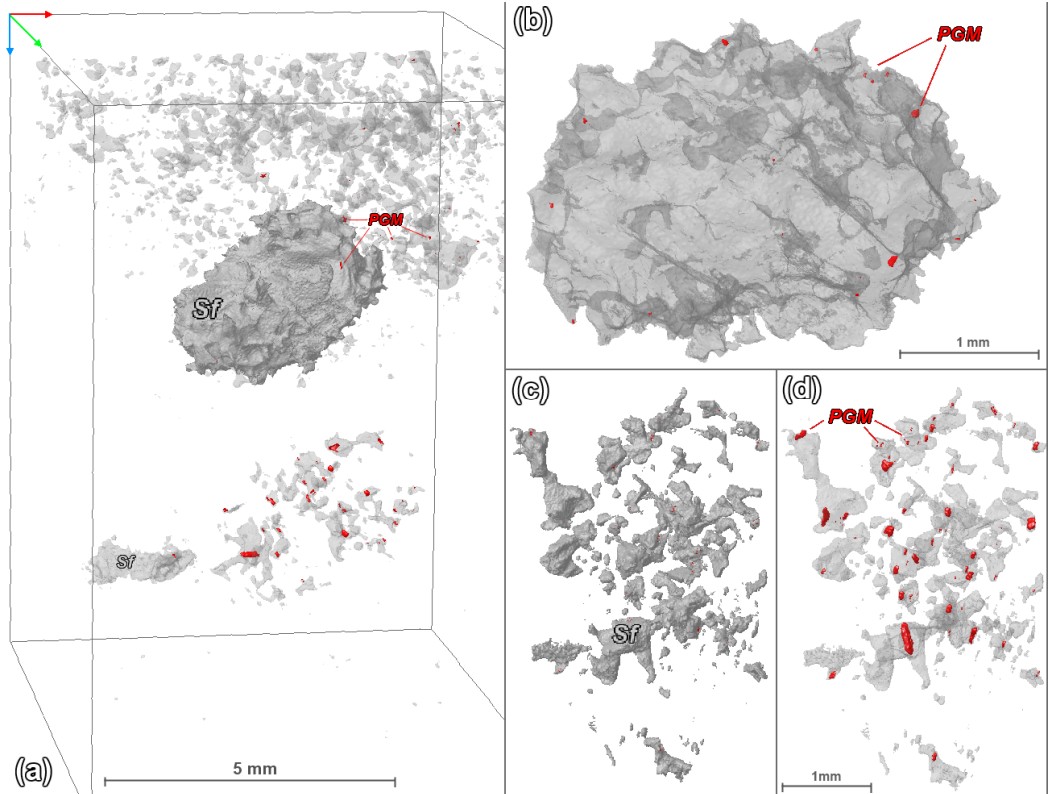

**Figure 4.** Results of microCT studies of the DV653-5 mineralized anorthosite: (**a**) volumetric reconstructions for a part of the 15 mm core, where sulfides (*Sf*) are shown in gray, the host silicate matrix is consistent with a transparent color, and p.PGMs are shown in red for clarity. The sulfide globule is shown opaque in this figure among other interstitial sulfides, including small grains of p.PGMs within a nest of scattered sulfides below the globule; (**b**) the spherical globule separately (here shown translucent); note that all p.PGMs are located at its surface as the silicate–sulfide boundary; and (**c,d**) sulfide nest anomalously enriched in p.PGMs in the opaque and translucent sulfide mode, respectively, indicating that all p.PGMs are located inside the sulfide. Video reconstructions are available at ref. 1 (https://www.youtube.com/watch?v=VVB16o5yfAk (accessed on 1 April 2024)) and ref. 2 (https://www.youtube.com/watch?v=3o08UnBMUlQ (accessed on 19 December 2024)).

As a result, it was concluded that potential PGMs (as the most X-ray contrast phases) in the volume of the 15 mm core are distributed unevenly; rare and relatively small (max. 35 μm, see Table 1) p.PGMs are found in the sulfides within the globule and on its periphery (see Figure 4b), with no signatures of any systematics. On the other hand, in the enriched sulfide nest, in which more than 90% (by volume) of all p.PGMs are concentrated, these grains can be divided according to two major criteria (see the captions to Figure 5), characterizing the following:

(1) Space relations between sulfides and rock-forming minerals attributed to (i) the tops of sulfide edges at the sulfide–silicate boundary, (ii) the triangular corners of interstices and segregations at the sulfide edges, (iii) a bridging in narrowing sulfide interstices, and (iv) rare grains within the sulfide that do not gravitate to the edges and corners;

(2) The p.PGM grain morphology visible as: (i) rounded, similar to a ball grain with a high sphericity coefficient (Spc) 0.85–0.96, usually relatively small in size (Figure 5f–l, small ones), (ii) flattened, sometimes slightly curved (Figure 5a,b) or elongated grains (Figure 5d–f,i,j) with Spc < 0.7, (iii) peanut-shaped paired aggregations of rounded or elongated grains, which can be small or quite large, up to 60–70 μm in size, and are the most common grains in the enriched nest (Figure 5i–l), and (iv) slightly elongated, egg-shaped, or oval p.PGM grains with Spc 0.72–0.85 (Figure 5c,g,h).

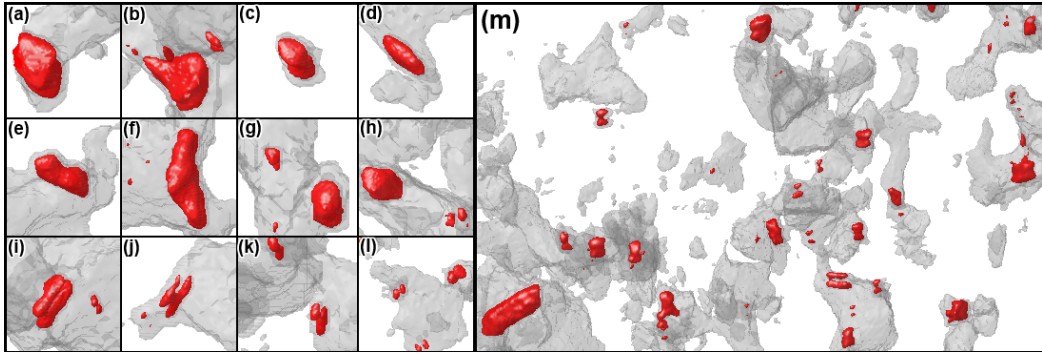

**Figure 5.** Examples of different morphology of the p.PGMs: (**a**–**l**) separate grains observed: (**d**,**e**,**g**,**h**,**j**,**k**) at the tops of sulfide edges at the sulfide–silicate boundary, (**a**,**b**,**g**,**h**,**l**) in triangular corners of interstices and segregations at sulfide edges, ((**f**) (coarse) and (**i**)) as sulfide bridging in tapering interstices, ((**c**,**g**), and (**f**) (fine)) inside sulfide, not gravitating to the edges and corners, and (**m**) general view of the p.PGM-enriched sulfide nest, including the largest elongated grain on the bottom left, which is of 450 μm in size. The scale in these figures is not given intentionally in order not to confuse the reader by projections of three-dimensional models on the screen plane.

## 4.2. The Average Sulfide Compositions

For microscopic studies of the sulfide globule, six tiny sections across its body were prepared. When calculating the "average sulfide", we used the same methodology as presented in [25]. Since the reflectivity of the main sulfide minerals is quite close to each other [26], each mineral phase was separated manually in the Paint. Net program. Then, for each processed panorama, the relative proportions of pyrrhotite (Po), pentlandite (Pn), cubanite (Cub), and chalcopyrite (Ccp) were calculated using the Adobe Photoshop CS2 program. As a result, the average for all the images corresponds to the following values: 39% Po, 21% Pn, 34% Cub, and 6% Ccp. This is consistent with approximately 35.2 wt.% S, 48.2% Fe, 6.4% Ni, 9.9% Cu, and 0.4% Co (Figure 6). The conversion to the average chemical composition was performed using average microprobe analyses for each sulfide mineral (Table 2).

We also calculated the average sulfide composition for the p.PGM-enriched sulfide nest (Figure 4c,d). This domain was studied on a separate sample, which, unlike the globule, was examined by the destructive method of the layer-by-layer sawing of the sample with step-by-step SEM and petrographic analysis for each polished section. A total of 10 saw cuts were made, primarily to study the compositions of p.PGMs. For each of the cuts, a corresponding panorama was visualized in the reflected light and processed according to the above-described actions. Thus, the following mean estimates (in vol.%) were obtained for the examined nest: Po—34, Pn—15, Ccp—23, and Cub—28, which are consistent with wt.% S—35.2, Fe—45.8, Ni—4.6, Cu—14.2, and Co—0.3 (Figure 6).

Thus, it was found that the PGM-enriched nest associated with the sulfide globule evidences a more Cu-rich bulk composition (14% vs. 10% Cu), relative to the globule (Figure 6), with the proportion of chalcopyrite increasing from 6% to 23%, so that the Ccp/Cub ratio increases from 0.17 to 0.82.

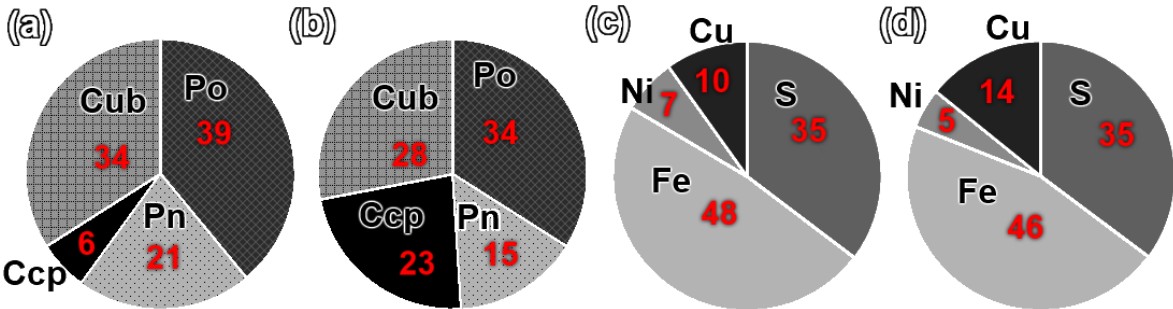

**Figure 6.** The mineral and chemical compositions of studied sulfide occurrences, including ratios of base metal sulfides and element concentrations: (**a**,**c**) in the globule, and (**b**,**d**) in the p.PGM-enriched nest.

**Table 2.** Compositions of base metal sulfides from 15 mm core (two analyses of each mineral are listed to show their insignificant range).

| Mineral | S (wt.%) | Fe | Co | Ni | Cu | Pt | Pd | Total | Formulae |
|---|---|---|---|---|---|---|---|---|---|
| *Po (Tr)* | 36.22 | 62.98 | 0.21 | bdl [1] | bdl | bdl | bdl | 99.41 | $Fe_{1.00}Co_{0.01}S$ |
| *Po (Tr)* | 36.23 | 63.55 | 0.33 | bdl | bdl | bdl | bdl | 100.11 | $Fe_{1.01}Co_{0.01}S$ |
| *Cub* | 35.15 | 40.95 | 0.14 | bdl | 22.75 | bdl | bdl | 98.99 | $Cu_{0.98}Fe_{2.01}Co_{0.01}S_3$ |
| *Cub* | 35.39 | 41.22 | 0.24 | bdl | 23.19 | bdl | bdl | 100.04 | $Cu_{0.99}Fe_{2.01}Co_{0.01}S_3$ |
| *Ccp* | 34.66 | 30.97 | 0.12 | bdl | 34.36 | bdl | bdl | 100.11 | $Cu_{1.00}Fe_{1.03}Co_{0.01}S_2$ |
| *Ccp* | 34.38 | 30.73 | 0.15 | bdl | 33.61 | bdl | bdl | 98.87 | $Cu_{0.99}Fe_{1.03}Co_{0.01}S_2$ |
| *Pn (glob)* | 32.99 | 35.3 | 0.79 | 29.87 | bdl | 0.18 | bdl | 99.13 | $(Ni_{3.96}Fe_{4.92}Co_{0.10})S_8$ |
| *Pn (glob)* | 32.92 | 35.56 | 1.09 | 29.81 | bdl | 0.23 | bdl | 99.61 | $(Ni_{3.96}Fe_{4.96}Co_{0.14})S_8$ |
| *Pn (nest)* | 32.8 | 35.15 | 0.87 | 30.13 | bdl | bdl | 1.23 | 100.18 | $(Ni_{4.00}Fe_{4.95}Co_{0.12})S_8$ |
| *Pn (nest)* | 33.02 | 35.12 | 0.94 | 30.23 | bdl | bdl | 0.89 | 100.2 | $(Ni_{4.00}Fe_{4.89}Co_{0.12})S_8$ |

[1] below detection limit.

### 4.3. SEM Studies and Mineralogy

As it was shown in the subchapter 4.2, the main globule is composed mainly of pyrrhotite/troilite, with the ratio of sulfur to iron being almost 1:1. All the troilite in the globule is cobalt-bearing (0.21–0.33 wt.%; average 0.28 wt.%). Cubanite, chalcopyrite, and pentlandite also contain noticeable amounts of Co: Cub 0.14–0.24 wt.% (on average 0.18 wt.%), Ccp 0.12–0.15 wt.% (0.13 wt.%), and Pn 0.79–1.09 wt.% (0.92 wt.%). Pentlandite from the globule also contains 0.22 wt.% Pt (on average, Table 2). Due to relatively high Co in all sulfide minerals, the "average sulfide" composition has a pretty high bulk Co concentration (0.4 wt.%, see 4.2). A similar pattern of Co distribution was observed in the PGM-enriched nest (Table 2), where the compositions of sulfide minerals do not differ from those determined for the globule. Only pentlandite contains an average of 1 wt.% Pd with no Pt (Table 2).

In the case of the sulfide globule, several PGM grains (up to the first tens of microns) were found at the sulfide–plagioclase boundary surface (Figure 7d,e): these are moncheite and tetraferroplatinum, as well as a phase of the composition $Pt_2Pd_2Sn$ (Table 3), which has been described in [21] as "unnamed $Pt_2Pd_2Sn$". In contrast, in the PGM-enriched nest (Figure 7c,f), which was opened by 10 layer-by-layer polishing, about 45 PGM grains were discovered and analyzed: 30 of them are tetraferroplatinum (PtFe), 8 are grains of potarite (PdHg), and 7 are grains of zvyagintsevite ($Pd_3Pb$) (Table 3).

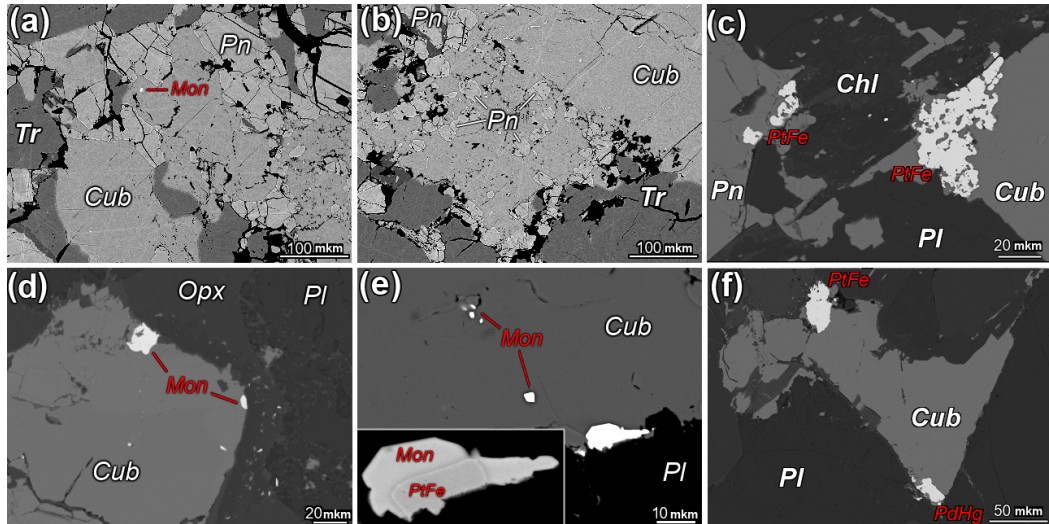

**Figure 7.** Mineralogical features of sulfide assemblages in the globule and associated PGM-enriched sulfide nest: (**a**) troilite (*Tr*), cubanite (*Cub*) (with micro-inclusion of moncheite—*Mon*), and pentlandite (*Pn*) in the sulfide globule; (**b**) pentlandite in cubanite at the boundary with troilite (in globule); (**c**) large sponge-like tetraferroplatinum grain in cubanite (on the *Sf-Pl* boundary, in close spatial association with chlorite in the PGM-enriched sulfide nest); (**d**) 25 μm moncheite grain in cubanite on the *Sf-Pl* boundary (globule); (**e**) 15 μm moncheite–tetraferroplatinum splicing in cubanite on the *Sf-Pl* boundary (globule); and (**f**) large grains of tetraferroplatinum and potarite in cubanite on the *Sf-Pl* boundary (PGM-enriched sulfide nest).

**Table 3.** Compositions of the PGMs encountered in the globule and in the enriched nest (2–3 analyses for each mineral are listed to show their extreme compositional scatter).

| Mineral | Pd (wt.%) | Sn | Te | Pt | Bi | Hg | Fe | Ni | Cu | Pb | Total |
|---|---|---|---|---|---|---|---|---|---|---|---|
| *Moncheite* | bdl [1] | bdl | 49.91 | 41.56 | 7.74 | bdl | bdl | bdl | bdl | bdl | 99.21 |
| *Moncheite* | bdl | bdl | 51.21 | 40.98 | 6.33 | bdl | bdl | bdl | bdl | bdl | 98.52 |
| *Moncheite* | bdl | bdl | 50.91 | 41.76 | 5.91 | bdl | bdl | bdl | bdl | bdl | 98.58 |
| *Unnamed $Pt_2Pd_2Sn$* | 28.43 | 21.8 | bdl | 48.93 | bdl | bdl | bdl | bdl | bdl | bdl | 99.16 |
| *Unnamed $Pt_2Pd_2Sn$* | 25.69 | 21.56 | bdl | 51.58 | bdl | bdl | bdl | bdl | bdl | bdl | 98.83 |
| *Tetraferroplatinum* | 4.05 | bdl | bdl | 82.65 | bdl | bdl | 11.59 | 0.48 | 1.27 | bdl | 100.04 |
| *Tetraferroplatinum* | 1.57 | bdl | bdl | 83.85 | bdl | bdl | 11.79 | 0.66 | 2.02 | bdl | 99.89 |
| *Tetraferroplatinum* | 1.31 | bdl | bdl | 74.66 | bdl | bdl | 14.58 | 0.94 | 7.5 | bdl | 98.99 |
| *Potarite* | 35.47 | bdl | bdl | bdl | bdl | 61.3 | 1.56 | bdl | bdl | bdl | 98.33 |
| *Potarite* | 33.42 | bdl | bdl | bdl | bdl | 62.56 | 0.51 | bdl | 0.67 | bdl | 97.16 |
| *Zvyagintsevite* | 57.08 | bdl | bdl | 1.18 | bdl | 1.55 | 0.49 | bdl | 0.54 | 35.66 | 96.5 |
| *Zvyagintsevite* | 58.04 | bdl | bdl | 1.04 | bdl | 2.64 | 0.24 | bdl | bdl | 35.44 | 97.4 |

[1] below detection limit.

Note that while tetraferroplatinum in the nest was found in 66% of the cases, it is presented mostly by large grains, and thus, by volume, constitutes about 90% of all the PGMs in the unusual domain. As can be seen in Figure 7c,f, the large grains of tetraferroplatinum (in contrast to the solid grains of the CT reconstructions, Figure 5), actually have a bunch-like, spongy structure. In this regard, it is quite difficult to correlate the morphology of PGMs with their composition, since the real details of PGMs morphology are not visualized on CT scans of the used resolution. The SEM studies also revealed a close association of the sulfide globule with large grains of orthopyroxene, as well as ilmenite and apatite. Apatite and ilmenite also occur in the PGM-enriched sulfide nest, and it contains 1–1.6 wt.% F and 2.4–2.6 wt.% Cl.

## 5. Discussion

The extremely heterogeneous pattern of the distribution of PGMs vs. sulfide material in the Dovyren Main Reef anorthosites was found for the first time. Its uniqueness is determined by three factors:

1. The first finding of a relatively large sulfide globule in the low-sulfide PGE-rich anorthosite. Despite the fact that smaller globules as much as 0.8–0.9 mm size are present in the 10-mm core (Figure 3), so far, globules larger than 1 mm have not been discovered in previous studies of the "reef" rocks. It is doubtful that their formation, to all occurrences, could be a result of a through pore migration of such a globule or smaller droplets of sulfides in the solidifying anorthositic matrix, which is difficult to imagine due to a very low permeability of the almost completely crystallized assemblage of the plagioclase grains. Instead, we suggest that the globule was formed in situ by the agglomeration of liquid sulfide microglobules disseminated in the intercumulus silicate melt, filling a large residual pore between the plagioclase crystals [3,8]. This is evidenced by the close association of this globule with a large orthopyroxene poikilitic crystal (see Figure 3e), occurring in the same space within the plagioclase matrix;

2. The Discovery of a localized domain, which looks like a nest of interstitial sulfides, in which the amount of PGMs in 100% sulfide volume exceeds 1.5 vol.%, whereas, in other sulfide occurrences, this concentration does not exceed 0.01 vol.%;

3. The combination of factors 1 and 2 suggests a non-random character of the co-existence of the relatively large but PGMs-poor sulfide spheroid and much smaller but anomalously enriched in PGM sulfide nest.

To better understand the contrast between the main sulfide globule and the sulfide-PGM nest, Table 1 displays data for two separate portions of the 15 mm core, which were divided virtually into two images—the globule alone (Figure 4b) (15 mm (A)) and the nest (Figure 4c,d) (15 mm (B)). The latter is only about 6 vol.% of the sulfides in the whole globule, but in terms of the 100% sulfide volume, the nest is 112 times more enriched in PGE!

So, the extreme enrichment of the sulfide nest in PGE-minerals is correlated to a marked shift in its average base metal composition towards a more Cu-rich sulfide material. Similar relations have been described in Ol-gabbronorite from an apophysis of the Dovyren intrusion, which contained even larger sulfide globules [25]. In the cited work, it was concluded that the occurrences of relatively Cu-rich and Cu-depleted sulfides in the same rock may be considered as a result of the differentiation of a protosulfide liquid, due to a late-stage migration of Cu-rich (post-MSS) sulfide residuals from the external boundary of the parental sulfide globule. We believe that similar processes of separation of residual sulfides could take place during the crystallization of an immiscible sulfide liquid initially originated within or close to the anorthositic agglomerations.

Considering probable mechanisms of such separation, one should take into account the high proportion of mercury in the studied PGMs (see Table 3), the close association of tetraferroplatinum grains with late hydrous minerals, such as chlorite (Figure 7e), as well as the presence of apatite in this sulfide domain, which manifest the probable role of late aqueous fluids in the formation of the finally observed noble-metal mineralization. Mass-balance calculations produce a very high Pt/Pd ratio in the PGE-rich sulfide nest of about 6, indicating an efficient fractionation between initial sulfide-controlled Pt and Pd. This separation may have occurred at the final stages of mineral-forming processes, charac-terized by the maximum role of late residual fluids. The mechanism of this fractionation, which led to a sharp increase in the Pt/Pd ratio in the enriched nest, may be attributed to a significant difference in the stability of the main aqueous hydrosulfide ($Pd(HS)_2$, $Pt(HS)_2$)

and chloride ($PdCl_4^{2-}$ and $PtCl_4^{2-}$) complexes, providing a much more efficient transfer of Pt from sulfides to the fluid phase as compared to Pd [27–29].

## 6. Conclusions

1. For the first time, a relatively large sulfide globule was found in the Main PGE-Reef of the Yoko-Dovyren massif. It has been established that the nest of disseminated sulfides associated with the globule is anomalously enriched in PGMs, with 90% of which being presented by tetraferroplatinum. As compared to 100% sulfide from the globule, this enrichment is 112-fold;

2. The average sulfide of the PGM-enriched nest is enriched in cuprous minerals, containing 28% Cub and 23% Ccp, while the main globule contained 34% and 6%, respectively. Accounting for the relative enrichment in PGMs, this may evidence a crystallization fractionation of a common sulfide precursor due to a mm-scale migration of the late Cu-enriched and probably PGE-rich sulfide residuals from the parental liquid to produce two observed types of sulfide material. This is supported by the difference in the PGE-mineralogy in the globule and in the nest: the former contains a few more high-temperature PGMs, such as moncheite, while the nest contains many low-temperature PGMs, including Pb- and Hg-bearing minerals, such as zvyagintsevite and potarite;

3. Despite the probable source of PGE being an immiscible sulfide liquid (primarily from the volume of the sulfide globule), late-stage water-bearing fluids could promote the redistribution of PGE during the separation of its crystallization products, being responsible for the extremely high Pt/Pd ratio of about 6 in the averaged sulfide nest. This is supported by the observed mineral association of tetraferroplatinum PtFe and Pb-Hg containing PGMs with chlorite and amphiboles in the presence of apatite. Experimental data on the differing stability of hydrosulfide and chloride complexes for Pt and Pd are consistent with this hypothesis [27–29].

**Author Contributions:** Conceptualization, I.V.P.; investigation, I.V.P. and D.V.K.; methodology, I.V.P.; resources, V.O.Y.; supervision, A.A.A.; validation, S.N.S. and G.S.N.; visualization, I.V.P.; writing—original draft, I.V.P.; and writing—review and editing, A.A.A. All authors have read and agreed to the published version of the manuscript.

**Funding:** The manuscript was supported by the RSF grant No 23-77-01036.

**Data Availability Statement:** The data supporting the reported results can be provided by the corresponding author upon request.

**Acknowledgments:** Initial versions of this paper were prepared within the scope of the government-financed research project FMUS-2019-0004 at the Vernadsky Institute of Geochemistry and Analytical Chemistry (RAS, Moscow). We would also like to thank E.V. Kislov for his assistance in the field-work and collection of samples, and three reviewers, whose comments improved the quality of the manuscript.

**Conflicts of Interest:** The authors declare no conflicts of interest.

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
