# Peer review of "Sulfide Globule and a Localized Domain Ultra-Enriched in PGMs in the Main Reef Anorthosite from the Yoko-Dovyren Massif"

_minerals, doi:10.3390/min15020160_

Round 1
Reviewer 1 Report
Comments and Suggestions for Authors
Dear Editors,
This is an interesting paper on the subject of extreme PGE enrichment in sulfides. I am happy to recommend it for publication after moderate revisions.
I have one principal question. The predominant phase in the nest is tetraferroplatinum (90 vol%), which, as the authors mention, formed during hydrothermal alteration of some earlier phases (as is well known for tetraferroplatinum in complexes of various types). However, Fig. 7e, showing a euhedral tetraferroplatinum crystal within moncheite, looks puzzling in this context. How did it form? For this reason, I would recommend dedicating more of the discussion to PGM alteration and the formation of related hydrous phases. Also, if the nests are 90% composed of isoferroplatinum, what is the Pt/Pd ratio? I presume it is very high. Was Pd removed during the hydrothermal stage, or was the sulfide initially enriched in Pt compared to Pd? Why was the late Cu-enriched sulfide liquid also so disproportionately enriched in Pt compared to Pd?
Please find my minor questions below.
Sections 1.2 and 1.3 belong to Methods.
Line 18: 3 m? Maybe, 3 mm?
Line 21: I suggest removing the exclamation mark.
Lines 56-57: Please mention the age, if possible.
Line 73: Please correct the spelling; it should be “Critical Zone.”
Line 124: What is “protosulfide precursor”? Perhaps “sulfide precursor” would be sufficient.
Line 136: Please ensure consistent citation style.
Line 155: This sentence is incomplete.
Lines 185-190: This section repeats content from section 1.3. Please consider merging the two.
Line 222: There is no clickable link visible; please ensure it is included in the final manuscript.
Line 223: I suggest removing the phrase, “This separation of the 15-mm core was not done by accident.”
Lines 232-240: This section describes the methodological differences between this study and the authors' previous work. I suggest rewriting it with the assumption that readers may not be familiar with the earlier paper. Focus exclusively on the new results without too much emphasis on the differences.
Line 282: The phrase “…to reliably separate them” is redundant, as it is already stated earlier in the same sentence.
Line 307: The phrase “or rather its extremely ferruginous variety–troilite” could be simplified to “or troilite.”
Lines 360-363: Is there any solid evidence to support this claim? Why are you confident that mm-sized sulfides could not migrate through plagioclase cumulate? This seems contradictory, as the next line mentions “large residual pores between plagioclase crystals.”
Line 371: The word “anusually” should be corrected to “unusually.”
Figure 3: The scale bar in panel “a” is barely visible. Additionally, the Cyrillic “м” is used instead of the Latin “m”; please correct this.
Reviewer 2 Report
Comments and Suggestions for Authors
Method 3D X-ray Computed Tomography (XCT) recent perspective path of the research distribution Sf and PGM minerals. But in most studies before this, no patterns were found in the distribution of sulfide minerals in the rock.
In this study, the fact of the division of sulfides into two areas with different composition and mineralogy was established for the first time.
In this article and previous works of these authors, the methodology and principles of mineral analysis are well substantiated.
The evolution of sulfide melts occurs with a parallel change in the composition of sulfide phases and, accordingly, it would be correct to indicate the evolution compositions of the sulphidic phases and in the nickel-rich cluster and the copper-rich cluster in the article.
Minor shortcomings of the analysis include the lack of information on the composition of sulfides in the first and second locations. Only for Pn in Table 2 can I see the composition in both locations. And results were similar. That is strange.
Also I see lost information in legends Fig 1-2. In first case Fig1(a) geological key N1-8 is not identified. In Fig 2 abbreviators of some minerals is not identified.
There is error on line 56. And “with schlierens and veins” (line 61) better – “schlieren structures and”.
Reviewer 3 Report
Comments and Suggestions for Authors
The manuscript under review is important and engaging. It examines patterns of platinum group metal concentration in rocks of layered intrusions with anorthosites of the Yoko-Dovyren massif as an example. The authors conducted a detailed study of sulfide globules using advanced high-resolution techniques, including X-ray microtomography (computed tomography, CT) and scanning electron microscopy (SEM). A contrasting behavior of PGE in anorthosites was identified, which the authors attribute to late-stage differentiation influenced by fluids.
Despite undeniable merits of the manuscript, the Reviewer has major concerns regarding both its structure and content.
Firstly, the opening part of the manuscript should be restructured. The "Introduction" section should not be divided into subsections. Here, the authors should clearly define the main objective of the study. Portions of the text from sections 1.1, 1.2, and 1.3 should be kept in the introduction and revised, while the parts describing the geological setting of the sulfide occurrences within the massif should be moved to a new section titled "Geological Background". Issues related to CT methodology are better suited for the "Materials and Methods" section.
Secondly, since the topic of the work is highly specific, the introduction should clarify a wider problem the authors address. In this regard, they should provide a brief literature review indicating the importance of tackling this issue. In addition, the introduction should include a concise overview of similar studies (CT) conducted on other layered intrusions.
Thirdly, the "Discussion" section lacks an in-depth analysis of similar studies carried out on other layered intrusions. What data were obtained in those studies? Were there similar contrasting distributions of PGE in sulfide segregations? If so, how were they explained? If not, this is especially important to highlight. It’s also worth considering whether contrasting distributions of PGE in sulfides were observed in former studies with no CT used. How did earlier researchers explain those findings and on what basis?
Answers to the above questions would not only strengthen the ground for the authors’ conclusions, but also expand the reference list, which is currently fairly limited (mainly containing studies directly related to the Yoko-Dovyren massif).
Additionally, the manuscript contains some technical flaws, such as subscripts in mineral formulas in Table 2 and typos in the references. The authors ensure these are corrected.
I believe that if the authors manage to revise the manuscript according to the above comments, it will be fully worthy of acceptance for publication.
Round 2
Reviewer 1 Report
Comments and Suggestions for Authors
Dear Authors and Editor,
I am satisfied with the way the Authors addressed my comments, and happy to recommend this paper for publication.
Author Response
Thank you for helpful comments and aprroving our reseach for the Minerals!
Reviewer 3 Report
Comments and Suggestions for Authors
The authors have done a great job refining the manuscript, and the paper is now fairly worth being published in Minerals.
There are, however, some minor points for consideration:
1. I recommend replacing “0” with “b.d.l.” (below detection limit) or a dash in Tables 2 and 3 (chemical analyses of minerals).
2. Since the journal welcomes full-color images, I recommend enhancing the readability of Figure 1 by replacing the map image from grayscale to color. It would also be beneficial to depict the host rocks in color (with a less saturated tone compared to the massif) to better highlight the boundaries, which are currently unclear in terms of what they delineate.
3. In Line 65, the authors mention “... size distribution (‘CSD’)”, but it is unclear for the reader what “C” stands for. This should be clarified.
4. In Table 1, the first line contains abbreviations that should be explained in a note, as they are not standard: conn.%, pics, lin.diam.
Once “um” apparently means micrometer, “u” should be replaced with “µ”.
5. The Reviewer would also like to draw the authors’ attention to the fact that the reference list contains 29 titles, 7 of which are the authors’ own works, representing 24%. While I was unable to find a specific guideline regarding the maximum allowance percentage of self-citations in the Author’s instructions, many journals typically limit this figure to around 15%. I would recommend that the authors make the appropriate adjustments.
Author Response
Comments 1: I recommend replacing “0” with “b.d.l.” (below detection limit) or a dash in Tables 2 and 3 (chemical analyses of minerals).
Response 1: Corrected
Comments 2: Since the journal welcomes full-color images, I recommend enhancing the readability of Figure 1 by replacing the map image from grayscale to color. It would also be beneficial to depict the host rocks in color (with a less saturated tone compared to the massif) to better highlight the boundaries, which are currently unclear in terms of what they delineate.
Response 2: Corrected
Comments 3: In Line 65, the authors mention “... size distribution (‘CSD’)”, but it is unclear for the reader what “C” stands for. This should be clarified.
Response 3: Corrected, see line 65
Comments 4: In Table 1, the first line contains abbreviations that should be explained in a note, as they are not standard: conn.%, pics, lin.diam. Once “um” apparently means micrometer, “u” should be replaced with “µ”.
Response 4: Corrected, see Table 1 and lines 229-230
Comments 5: The Reviewer would also like to draw the authors’ attention to the fact that the reference list contains 29 titles, 7 of which are the authors’ own works, representing 24%. While I was unable to find a specific guideline regarding the maximum allowance percentage of self-citations in the Author’s instructions, many journals typically limit this figure to around 15%. I would recommend that the authors make the appropriate adjustments.
Response 5: We have reduced the number of citations of co-authors of the manuscript from 7 to 5. It is maximum what we can do, as most cited results on the Dovyren intrusion were obtained exclusively by our team. Hope for your understanding.